# Metaplot: A new Stata module for assessing heterogeneity in a meta-analysis

**Jalal Poorolajal**[1,2]*, **Shahla Noornejad**[3]

**1** Department of Epidemiology, School of Public Health, Hamadan University of Medical Sciences, Hamadan, Iran, **2** Modeling of Noncommunicable Diseases Research Center, School of Public Health, Hamadan University of Medical Sciences, Hamadan, Iran, **3** School of Public Health, Hamadan University of Medical Sciences, Hamadan, Iran

* poorolajal@umsha.ac.ir

**Data Availability Statement:** All relevant data are within the manuscript and its Supporting information files.

## Abstract

### Background

The proposed sequential and combinatorial algorithm, suggested as a standard tool for assessing, exploring, and reporting heterogeneity in the meta-analysis, is useful but time-consuming particularly when the number of included studies is large. Metaplot is a novel graphical approach that facilitates performing sensitivity analysis to distinguish the source of substantial heterogeneity across studies with ease and speed.

### Method

Metaplot is a Stata module based on Stata's commands, known informally as "ado". Metaplot presents a two-way (x, y) plot in which the x-axis represents the study codes and the y-axis represents the values of $I^2$ statistics excluding one study at a time (n-1 studies). Metaplot also produces a table in the 'Results window' of the Stata software including details such as $I^2$ and $\chi^2$ statistics and their *P*-values omitting one study in each turn.

### Results

Metaplot allows rapid identification of studies that have a disproportionate impact on heterogeneity across studies, and communicates to what extent omission of that study may reduce the overall heterogeneity based on the $I^2$ and $\chi^2$ statistics. Metaplot has no limitations regarding the number of studies or types of outcome data (binomial or continuous data).

### Conclusions

Metaplot is a simple graphical approach that gives a quick and easy identification of the studies having substantial influences on overall heterogeneity at a glance.

**Funding:** The Vice-Chancellor of Research and Technology, Hamadan University of Medical Sciences funded this study (No. 9603161751). However, the funder had no role in the study design, data collection and analysis, decision to publish, or preparation of the manuscript.

**Competing interests:** The authors have declared that no competing interests exist.

## Introduction

The studies that are brought together in a meta-analysis inevitably differ in many aspects. This variability across studies is called heterogeneity [1]. The between-studies heterogeneity can be assessed by the chi-square test also written as $\chi^2$ or Chi$^2$ and can be quantified by $I^2$ statistics [2, 3]. When there is heterogeneity in a meta-analysis, the source of heterogeneity across studies should be carefully investigated on a case-by-case basis [4].

A common approach, which was proposed by Patsopoulos et al, is to perform a sensitivity analysis based on a sequential and combinatorial algorithm [5]. According to this algorithm, one study is excluded from the meta-analysis at a time and the impact of the excluded study on the between-study heterogeneity is evaluated based on $I^2$ statistic and $\chi^2$ test. This 'one-out' sensitivity analysis tells us to what extent the overall heterogeneity changes by excluding a particular study at a time. Then, the study that is responsible for the largest decrease in $I^2$ value should be dropped out. This process is repeated for a new set of n-1 studies. This sequential and combinatorial algorithm is repeated several times until the $I^2$ statistic drops below the desired threshold value of 50%. In the last step, there is a possibility that more than one omitted study can result in $I^2$ dropping below the intended threshold. In such cases, the algorithm that results in the maximum decrease in the $I^2$ statistic below the desired threshold is selected. There is a chance that two or more studies cause the same reduction in $I^2$ by their exclusion. In this case, the study with the largest reduction in $\chi^2$ statistic (the least $\chi^2$ statistic) is dropped out.

Based on the aforementioned algorithm, this 'one-out' sensitivity analysis must be repeated n-1 times to specify and exclude the outlying study from the meta-analysis. If the desired threshold value of 50% is not achieved in the first step, the algorithm must be repeated n-2, n-3, etc. Therefore, this algorithm may be boring and time-consuming when the number of included studies is large and the between-studies heterogeneity is substantial.

In this study, we aimed to introduce a novel Stata graph that performs the 'one-out' sensitivity analysis for n-1 studies and identifies immediately the studies responsible for substantial heterogeneity across studies by executing "metaplot.ado" Stata command.

## Methods

Metaplot is a Stata module based on Stata's commands, known as "ado". Metaplot produces a two-dimensional (x, y) Stata graph. The x-axis represents the included studies. The studies are shown on this axis by an ID code. The y-axis represents the values of $I^2$ statistics based on 'one-out' (n-1 studies) sensitivity analysis indicating to what extent the overall heterogeneity changes by excluding a particular study at a time.

Furthermore, the "metaplot" command generates a table in the "Results window" of the Stata including more details about 'one-out' sensitivity analysis in terms of the $I^2$ and $\chi^2$ statistics and their *P*-values. In addition to study codes, the studies' identifications can be presented in the table.

The "metaplot" command is flexible and works with any measurement option including binary data (effect size + standard error or effect size + confidence intervals) and continuous data (sample + mean + standard deviation). The full form of the "metaplot" command is as follows

```
metaplot varlist [if] [in] [, id(study) tr(#)]
```
where

- "varlist" can be "a b c d" or "lnes se" or "es lles ules" or "n1 mean1 sd1 n0 mean0 sd0"

- "id(study)" option displays studies identifications (the first authors and the year of publication) specified by the variable "study" in the dataset.

- "tr(#)" option specifies the desired threshold values for example: 0.4, 0.5, 0.6, 0.65, 0.8, etc.

  The abbreviations in the above command represent the following terms.

- "a b c d" represents "events" and "non-events" in the intervention (exposure) and control groups, respectively.

- "lnes" represents the "Naperian logarithm" of the effect size that may be risk ratio (lnrr) or odds ratio (lnor).

- "se" represents the standard error of the effect size.

- "es" represents the effect size that may be risk ratio (rr) or odds ratio (or).

- "lles" represents the lower limit of the confidence interval for the effect size.

- "ules" represents the upper limit of the confidence interval for the effect size.

- "n1" and "n0" represent the sample size for the intervention (exposure) and control groups, respectively.

- "mean1" and "mean0" represent the mean for the intervention (exposure) and control groups, respectively.

- "sd1" and "sd0" represent the standard deviation for the intervention (exposure) and control groups, respectively.

  The relevant files including "metaplot.ado" and "metaplot.hlp" are attached to this paper as (S1 and S2 Files).

## Results

To show the capability and flexibility of the 'metaplot" command we used various datasets (S1–S3 Datasets) related to our previous published meta-analyses [6–8].

The first dataset (S1 Dataset), which was used to introduce the "metaplot" module, related to a published meta-analysis addressed the risk factors for stomach cancer [6]. This is a dataset with a "binomial" outcome (stomach cancer). In this meta-analysis, 15 studies addressed the association between stomach cancer and drinking black tea. The heterogeneity across studies was high ($I^2$ = 64.23%). To perform sensitivity analysis using the "metaplot" command for this dataset, we executed the following command in the Stata software.

- metaplot es lles ules, id(study)

  The result of the above command is given in Fig 1. This figure shows the results of the 'one-out' sensitivity analysis using the "metaplot" command. According to this figure, all values of $I^2$ statistics excluding one study at a time (n-1 studies) were above the desired threshold value of 50% except for study #5. By omitting study #5 from the meta-analysis, the heterogeneity fell below the desired threshold value of 50%. That means this study was an outlier and the main reason for heterogeneity across studies. Table 1 shows the results of 'one-out' sensitivity analysis in detail including $I^2$ and $\chi^2$ statistics and their *P*-values omitting one study at a time. Based on this table, the overall heterogeneity across studies was high ($I^2$ = 64.23%). However, the heterogeneity decreased to 38.93% after omitting study #5.

  The second dataset (S2 Dataset), which was used to introduce the "metaplot" module, related to a published meta-analysis addressed the effect of oral potassium supplementation on

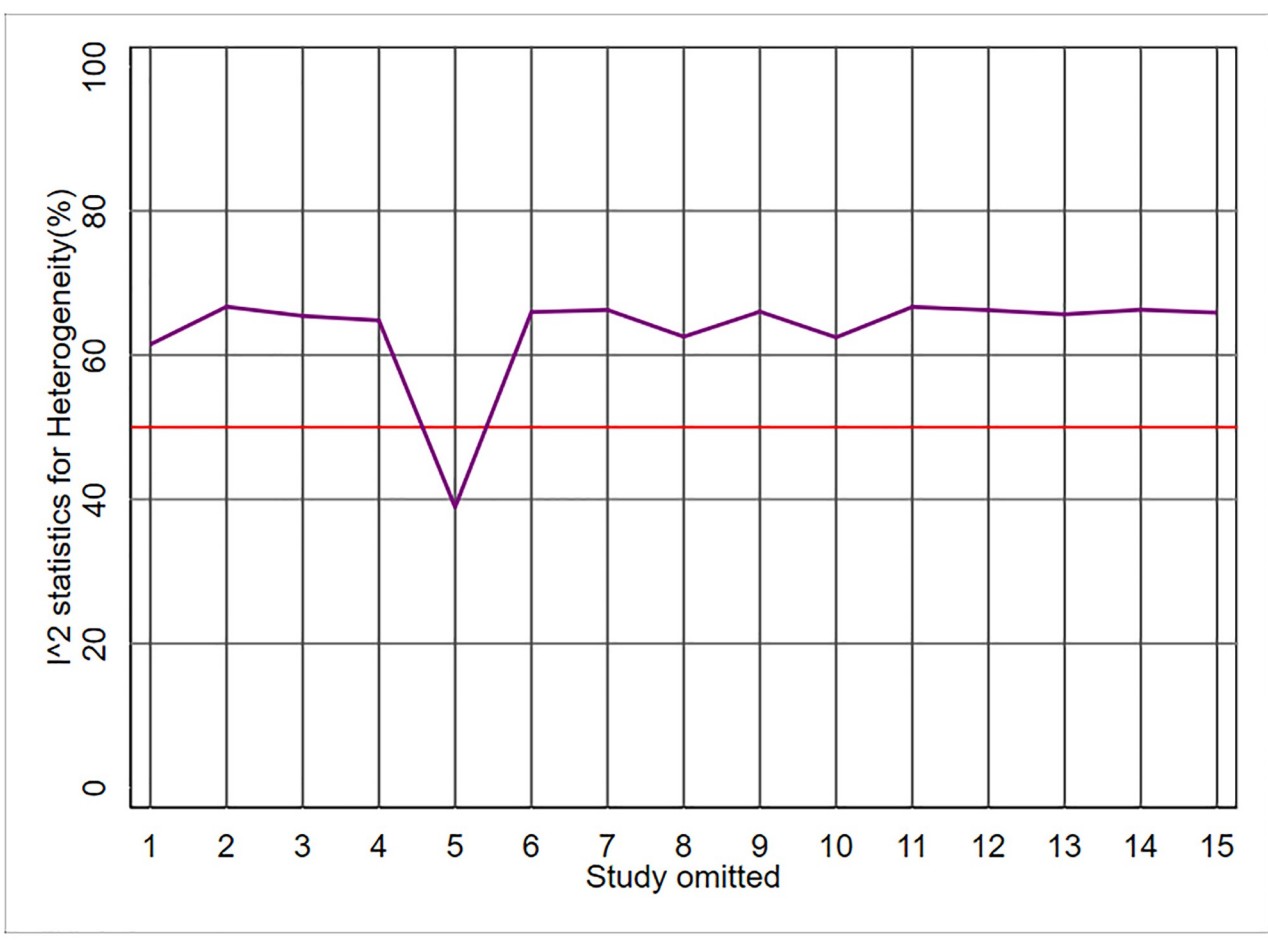

**Fig 1. Meta-analyses of risk factors for stomach cancer; metaplot delineates I$^2$ statistics and $\chi^2$ statistics and their *P*-values based on 'one-out' sensitivity analysis [Stata command: Metaplot es lles rules, id(study)].**

the management of essential hypertension [7]. This is a dataset with a "continuous" outcome (blood pressure). In this meta-analysis, 22 studies addressed the effect of oral potassium supplementation on diastolic blood pressure. The heterogeneity across studies was high (I$^2$ = 81.88%). To perform sensitivity analysis using the "metaplot" command for this dataset, we executed the following command in the Stata software.

- metaplot n1 mean1 sd1 n0 mean0 sd0, id(study)

The result of the above command is given in Fig 2. This figure shows the results of the "metaplot" command based on a 'one-out' sensitivity analysis. According to this figure, all values of I$^2$ statistics excluding one study at a time (n-1 studies) were above the desired threshold value of 50%. However, the effect of omitting one study at a time was not similar across studies. For example, studies #14, #3, and #5 were responsible for the largest decrease in I$^2$ values, respectively. Although heterogeneity decreased significantly, particularly by omitting study #14, it did not reach below the threshold value of 50%. Therefore, this process should be repeated for a new set of n-1 studies after omitting study #14. According to the results of Table 2, the overall heterogeneity across studies was high (I$^2$ = 81.88%). However, the heterogeneity decreased to 67.76%, 75.19%, and 79.85% after omitting studies #14, #3, and #5, respectively.

**Table 1. Meta-analyses of risk factors for stomach cancer; results of "metaplot" command.**

| Study omitted | I2 | [95% Conf. Interval] | | Chi2 | P>|t| |
|---|---|---|---|---|---|
| 1 Baroudi 2014 | 61.48 | 31.09 | 78.46 | 33.75 | 0.001 |
| 2 Takezaki 2001 | 66.71 | 41.62 | 81.02 | 39.05 | 0.000 |
| 3 Goldbohm 1996 | 65.43 | 39.06 | 80.39 | 37.60 | 0.000 |
| 4 Gallus 2009 | 64.81 | 37.83 | 80.09 | 36.95 | 0.000 |
| 5 Chew 1999 | 38.93 | 0.00 | 67.60 | 21.29 | 0.067 |
| 6 Watabe 1998 | 65.98 | 40.16 | 80.66 | 38.21 | 0.000 |
| 7 Inoue 1994 | 66.26 | 40.72 | 80.80 | 38.53 | 0.000 |
| 8 Hoshiyama 1992 | 62.55 | 33.27 | 78.98 | 34.71 | 0.001 |
| 9 Al-qadasl 2016 | 66.03 | 40.25 | 80.68 | 38.26 | 0.000 |
| 10 Hansson 1993 | 62.46 | 33.08 | 78.94 | 34.63 | 0.001 |
| 11 Galanis 1998 | 66.68 | 41.57 | 81.01 | 39.02 | 0.000 |
| 12 Chen 2009 | 66.23 | 40.67 | 80.78 | 38.50 | 0.000 |
| 13 Inoue 1998 | 65.65 | 39.50 | 80.50 | 37.85 | 0.000 |
| 14 Bao 2004 | 66.29 | 40.78 | 80.81 | 38.57 | 0.000 |
| 15 La Vecchia 1992 | 65.88 | 39.96 | 80.61 | 38.10 | 0.000 |
| **Combined** | **64.23** | **37.88** | **79.40** | **39.14** | **0.000** |

The third dataset (S3 Dataset), which was used to introduce the "metaplot" module, related to a published meta-analysis addressed the preventable factors for primary prevention of childhood obesity [8]. This is a dataset with a "binomial" outcome (stomach cancer) and multiple studies. In this meta-analysis, 84 studies addressed the association between physical activity and childhood obesity. The heterogeneity across studies was high ($I^2$ = 96%). We used the sequential and combinatorial algorithm and performed a 'one-out' sensitivity analysis and repeated the process several times. For this purpose, we executed the following command in the Stata software for n-1 studies several times.

- metaplot lnor se, id(study)

The result of the above command is given in Fig 3. This figure shows the last step when the $I^2$ statistic dropped below the desired threshold value of 50% by omitting just one more study. By looking at Fig 3 one can realize that there are at least 5 options to reduce the $I^2$ statistic below the value of 50%. By omitting any of the studies #13, #16, #25, #37, and #57 the $I^2$ statistic drops below the value of 50% and reaches 49.25%, 48.35%, 49.95%, 49.16%, and 47.25%, respectively (Table 3). When there is a possibility that more than one omitted study can result in $I^2$ dropping below the intended threshold, the study that results in the maximum decrease in the $I^2$ statistic below the desired threshold is selected. Accordingly omitting study #57 is the best choice. There might have been a chance that two or more studies caused the same reduction in $I^2$ by their exclusion. In that case, the study with the largest reduction in $\chi^2$ statistic (the least $\chi^2$ statistic) would have been dropped out.

## Discussion

The idea of Metaplot, which was first introduced in 2010 [9], is a simple graphical approach to identify outliers and their effects on overall heterogeneity across studies. Patsopoulos et al. [5] suggested the sequential and combinatorial algorithm for performing sensitivity analyses. This algorithm is a useful method for assessing, exploring, and reporting the between-study heterogeneity in the meta-analysis but is time-consuming when the number of included studies is large and heterogeneity is substantial. For example, as noted in the results section, 84 studies

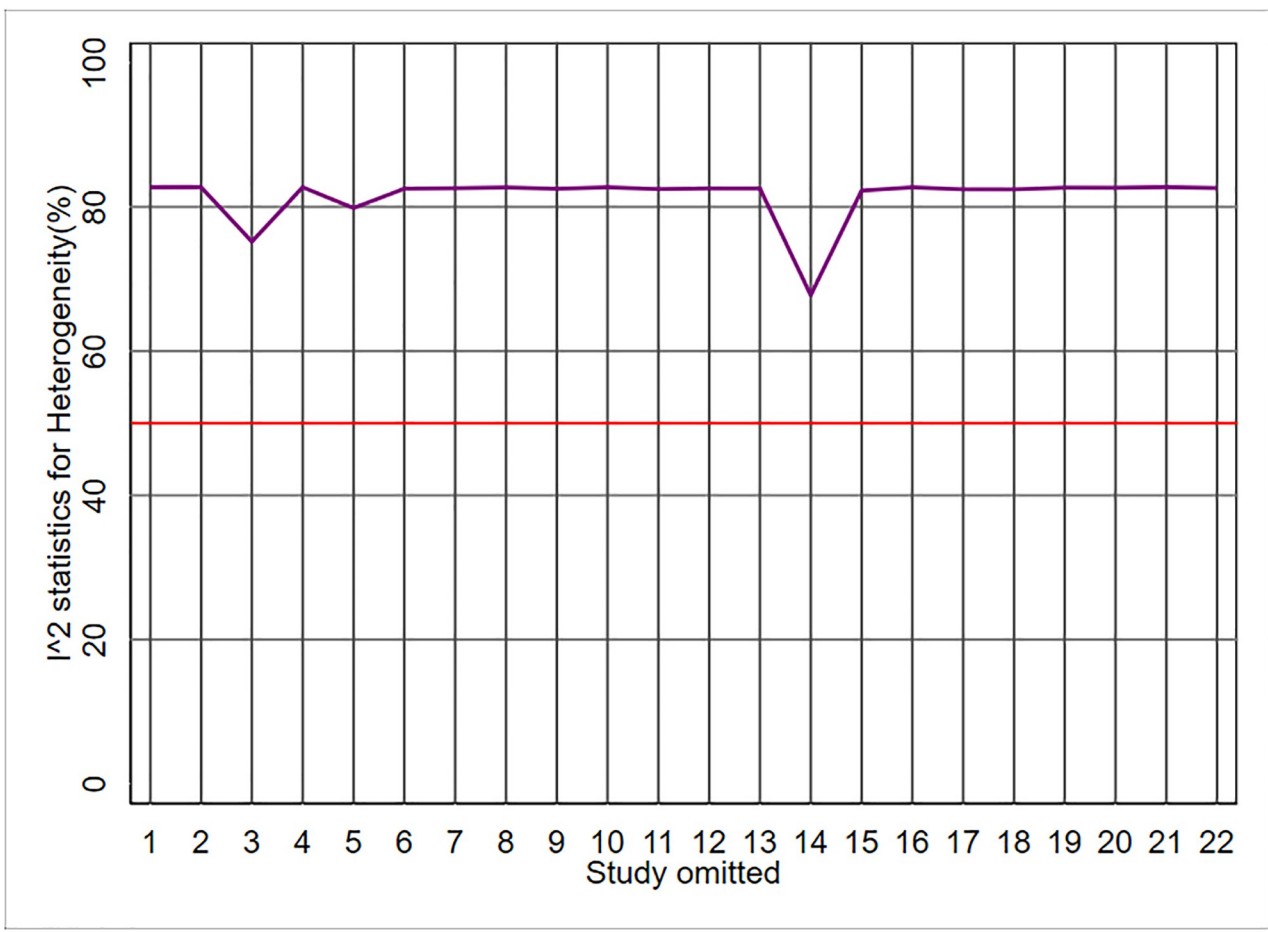

**Fig 2. Meta-analyses of oral potassium supplementation for the management of essential hypertension; metaplot delineates $I^2$ statistics and $\chi^2$ statistics and their *P*-values based on 'one-out' sensitivity analysis [Stata command: Metaplot n1 mean1 sd1 n0 mean0 sd0, id(study)].**

addressed the association between physical activity and childhood obesity [8]. In this case, the sequential and combinatorial algorithm needs to be repeated hundreds of times particularly when the heterogeneity across studies is substantial. While by executing the "metaplot" command we can perform 'one-out' sensitivity analysis across several studies, no matter how many they are, and identify immediately to what extent the overall heterogeneity changes by excluding a particular study at a time. Another capability of the "metaplot" command is its flexibility. It is possible to execute this command for meta-analysis of different types of outcome data (e.g. binary, continuous, or time to event) and different types of summary measures (e.g. odds ratio, risk ratio, rate ratio, or hazard ratio).

The $I^2$ threshold value of 50% usually depends on the type of research we are performing. The threshold value of 50% is not rigid in the "metaplot" command. A rigid threshold value for the interpretation of $I^2$ can be misleading since the importance of inconsistency depends on several factors [1]. The "metaplot" command has the option "tr(#)" that establishes different threshold values.

Care must be taken in the interpretation of the chi-squared test since it has low power in the situation of a meta-analysis when studies have a small sample size or are few in number. This means that while a statistically significant result may indicate a problem with

**Table 2. Meta-analyses of oral potassium supplementation for the management of essential hypertension; results of "metaplot" command.**

| Study omitted | I2 | [95% Conf. Interval] | | Chi2 | P>\|t\| |
|---|---|---|---|---|---|
| 1 Forrester 1988 | 82.72 | 74.64 | 88.23 | 115.74 | 0.000 |
| 2 Fotherby 1992 | 82.73 | 74.66 | 88.23 | 115.83 | 0.000 |
| 3 Franzoni 2005 | 75.19 | 62.10 | 83.75 | 80.60 | 0.000 |
| 4 Gijsbers 2015 | 82.73 | 74.65 | 88.23 | 115.79 | 0.000 |
| 5 Grimm 1988 | 79.85 | 69.93 | 86.49 | 99.24 | 0.000 |
| 6 Grobbee 1987 | 82.53 | 74.33 | 88.11 | 114.47 | 0.000 |
| 7 He 2010 | 82.60 | 74.44 | 88.15 | 114.92 | 0.000 |
| 8 Heseltine 1990 | 82.70 | 74.61 | 88.21 | 115.61 | 0.000 |
| 9 Kaplan 1985 | 82.51 | 74.30 | 88.10 | 114.37 | 0.000 |
| 10 Kawano 1998 | 82.72 | 74.64 | 88.23 | 115.75 | 0.000 |
| 11 Lawton 1990 | 82.47 | 74.23 | 88.07 | 114.09 | 0.000 |
| 12 MacGregor 1982 | 82.56 | 74.38 | 88.13 | 114.67 | 0.000 |
| 13 MacGregor 1984 | 82.56 | 74.38 | 88.13 | 114.67 | 0.000 |
| 14 Patki 199076 | 67.76 | 49.27 | 79.51 | 62.04 | 0.000 |
| 15 Rahimi 2007 | 82.25 | 73.88 | 87.94 | 112.71 | 0.000 |
| 16 Richards 1984 | 82.71 | 74.63 | 88.22 | 115.70 | 0.000 |
| 17 Siani 1987 | 82.43 | 74.17 | 88.05 | 113.82 | 0.000 |
| 18 Siani 1991 | 82.43 | 74.16 | 88.05 | 113.80 | 0.000 |
| 19 Smith 1985 | 82.67 | 74.56 | 88.19 | 115.40 | 0.000 |
| 20 Svetkey 1987 | 82.66 | 74.54 | 88.19 | 115.32 | 0.000 |
| 21 Valdes 1991 | 82.74 | 74.67 | 88.24 | 115.87 | 0.000 |
| 22 Wu 200682 | 82.62 | 74.47 | 88.16 | 115.05 | 0.000 |
| **Combined** | 81.88 | 73.51 | 87.6 | 115.88 | **0.000** |

heterogeneity, a non-significant result must not be taken as evidence of no heterogeneity [1]. This is also why a P-value of 0.10 is sometimes used, rather than the conventional level of 0.05. Another problem with the test is that when there are many studies in a meta-analysis, the test has a high power to detect a small amount of heterogeneity that may be clinically unimportant.

Huedo-Medina et al. [10] examined and compared the performances of the Q test and the $I^2$ index for assessing homogeneity across individual studies in meta-analysis. They confirmed that the Q test only reports the presence or absence of homogeneity across studies but does not specify the extent of such heterogeneity. On the other hand, the $I^2$ index can quantify the degree of heterogeneity. Although the $I^2$ index has the same problems of low statistical power with a small number of studies, they suggested the $I^2$ index as a complement to the Q test.

The raw idea of "metaplot" was first introduced in 2010 [9]. This preliminary idea was never implemented actually at that time because the package had not been generated yet. The new design of the "metaplot" presented in this paper is very different from the original one introduced in 2010. The original design was a complicated three-dimensional graph with x, y, and z axes including unnecessary information. It was rather hard to understand. The new design of "metaplot" is a two-dimensional graph with x and y axes. Furthermore, we added a table including details of information ($I^2$ and $\chi^2$ statistics and their $P$-values omitting one study in each turn) to simplify the interpretation of the 'metaplot' graph. In the current paper, we explained the capability of the "Metaplot" module and how to use the Stata command and its options. We examined this module on different real datasets and reported the results.

There are several graphical methods for the exploration of heterogeneity in the meta-analysis. One of these methods is the traditional Galbraith plot [11, 12]. This plot provides a

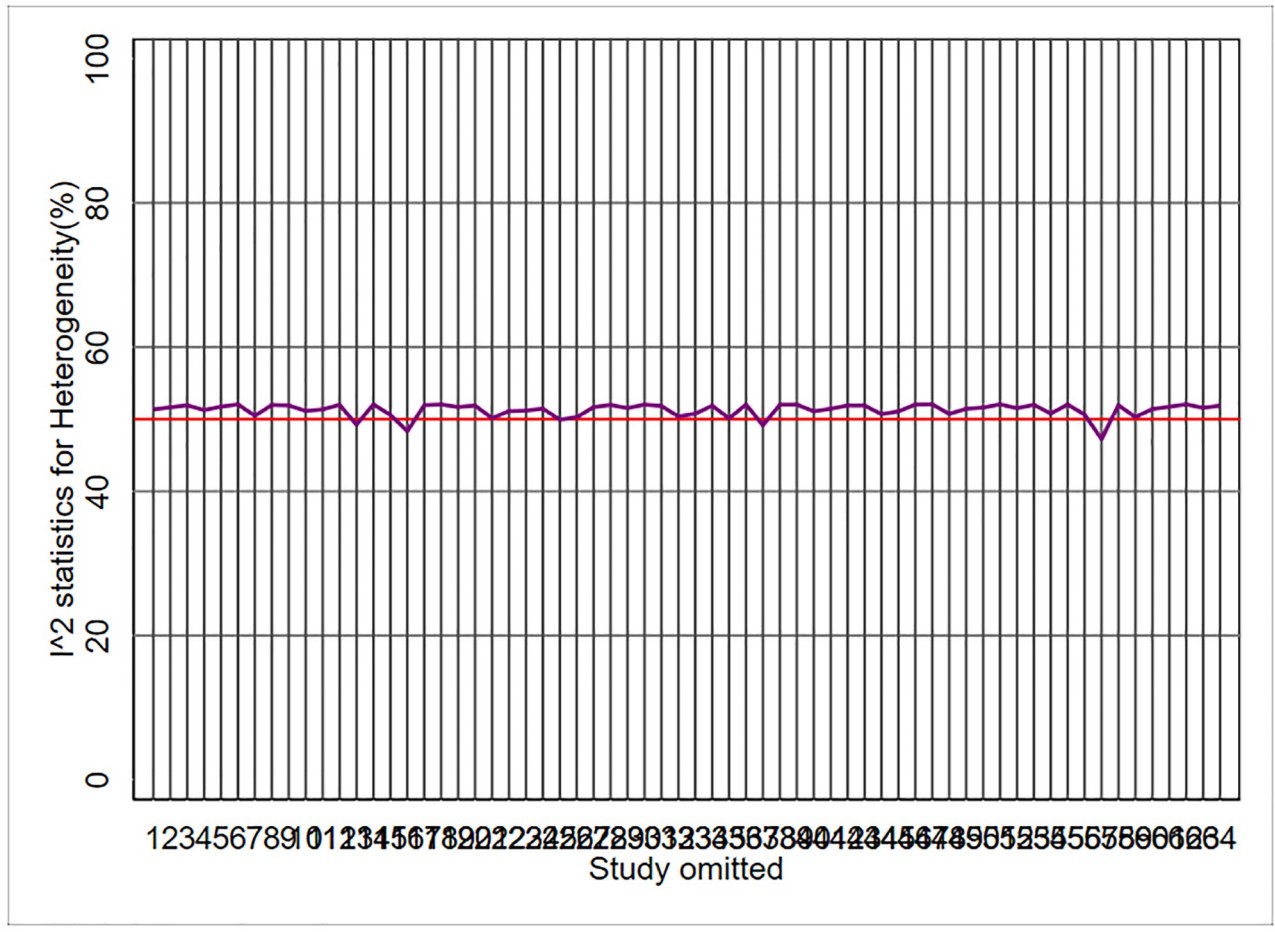

**Fig 3. Meta-analyses of primary prevention of childhood overweight and obesity by preventable behavioral factors; metaplot delineates I² statistics and χ² statistics and their *P*-values based on 'one-out' sensitivity analysis [Stata command: Metaplot lnor se, id(study)].**

graphical display to get a visual impression of the amount of heterogeneity from a meta-analysis. For each study, the observed effect sizes on the vertical axis are plotted against the reciprocal standard errors on the horizontal axis. The regression line projects through the origin, with its 95% confidence interval positioned 2 units over and below the regression line, has a slope equal to the overall log rate ratio. In the absence of heterogeneity, we could expect all the points to lie within the confidence bounds. The L'Abbé plot is another useful method for assessing heterogeneity in the meta-analysis [13, 14]. It is a scatter plot with the risk in the control group on the x-axis and the risk in the experimental group on the y-axis. The visual inspection gives a quick and easy indication of the studies having different results from other studies. These studies are considered outliers and hence potential sources of heterogeneity. Although these graphical procedures are useful and their interpretations are straightforward, they have a major limitation. When only one study causes extreme heterogeneity, these methods point to the same study as Metaplot suggests. However, in situations where the heterogeneity is resulted from several studies, the above graphical procedures are impractical to indicate to what extent a particular study influences the overall heterogeneity. Our proposed graphical method has overcome this problem. According to Metaplot method, one study is excluded from the meta-analysis at a time and the impact of the excluded study is evaluated on the overall

**Table 3. Meta-analyses of primary prevention of childhood overweight and obesity by preventable behavioral factors; results of "metaplot" command.**

| Study omitted | I2 | [95% Conf. Interval] | | Chi2 | P>t |
|---|---|---|---|---|---|
| 1 Adachi-Mejia 2007 | 51.36 | 35.01 | 63.59 | 127.46 | 0.000 |
| 2 Al-Domi 2019 | 51.65 | 35.43 | 63.79 | 128.22 | 0.000 |
| 3 Al-Hazzaa 2012 | 51.93 | 35.83 | 63.99 | 128.98 | 0.000 |
| 4 Al-Muhaimeed 2015 | 51.27 | 34.88 | 63.53 | 127.23 | 0.000 |
| 5 Arango 2011 | 51.74 | 35.56 | 63.86 | 128.47 | 0.000 |
| 6 Badr 2017 | 52.06 | 36.01 | 64.08 | 129.32 | 0.000 |
| 7 Basterfield 2014 | 50.47 | 33.74 | 62.98 | 125.17 | 0.000 |
| 8 Bhuiyan 2013 | 51.97 | 35.89 | 64.02 | 129.09 | 0.000 |
| 9 Bibiloni 2010 | 51.91 | 35.80 | 63.97 | 128.92 | 0.000 |
| 10 Boričić 2014 | 51.15 | 34.71 | 63.45 | 126.92 | 0.000 |
| 11 De Lucinéia 2014 | 51.36 | 35.02 | 63.60 | 127.48 | 0.000 |
| 12 Dudas 2008 | 52.00 | 35.93 | 64.03 | 129.15 | 0.000 |
| 13 Duncan 2011 | 49.25 | 31.99 | 62.13 | 122.16 | 0.000 |
| 14 Dupuy 2011 | 52.06 | 36.02 | 64.08 | 129.33 | 0.000 |
| 15 Eker 2018 | 50.59 | 33.91 | 63.06 | 125.48 | 0.000 |
| 16 Fu 2004 | 48.35 | 30.70 | 61.51 | 120.04 | 0.000 |
| 17 Gharib 2008 | 51.94 | 35.85 | 64.00 | 129.01 | 0.000 |
| 18 Ghosh 2015 | 52.04 | 35.99 | 64.07 | 129.28 | 0.000 |
| 19 Godakanda 2018 | 51.69 | 35.49 | 63.82 | 128.34 | 0.000 |
| 20 Ha 2005 | 51.88 | 35.77 | 63.96 | 128.86 | 0.000 |
| 21 Hajian-Tilaki 2012 | 50.15 | 33.28 | 62.76 | 124.38 | 0.000 |
| 22 Haug 2009 | 51.10 | 34.65 | 63.41 | 126.80 | 0.000 |
| 23 Honório 2014 | 51.19 | 34.77 | 63.47 | 127.01 | 0.000 |
| 24 Januszek-Trzciakowska 2014 | 51.46 | 35.16 | 63.67 | 127.74 | 0.000 |
| 25 Keane 2017 | 49.95 | 32.99 | 62.61 | 123.87 | 0.000 |
| 26 Kuhle 2010 | 50.28 | 33.47 | 62.84 | 124.70 | 0.000 |
| 27 Leatherdale 2013 | 51.68 | 35.47 | 63.81 | 128.31 | 0.000 |
| 28 Liu 2012 | 51.98 | 35.90 | 64.02 | 129.11 | 0.000 |
| 29 Lowry 2012 | 51.54 | 35.27 | 63.72 | 127.93 | 0.000 |
| 30 Lätt 2015 | 52.02 | 35.96 | 64.05 | 129.23 | 0.000 |
| 31 Macwana 2017 | 51.84 | 35.70 | 63.92 | 128.73 | 0.000 |
| 32 Mahfouz 2011 | 50.38 | 33.61 | 62.91 | 124.95 | 0.000 |
| 33 Mansoori 2018 | 50.75 | 34.15 | 63.17 | 125.90 | 0.000 |
| 34 Melkevik 2015 | 51.87 | 35.75 | 63.95 | 128.83 | 0.000 |
| 35 Muntaner-Mas 2017 | 50.12 | 33.23 | 62.73 | 124.29 | 0.000 |
| 36 Mushtaq 2011 | 52.04 | 35.98 | 64.06 | 129.26 | 0.000 |
| 37 Nasreddine 2014 | 49.16 | 31.86 | 62.07 | 121.95 | 0.000 |
| 38 Neutzling 2003 | 52.02 | 35.96 | 64.05 | 129.22 | 0.000 |
| 39 Oellingrath 2017 | 52.02 | 35.97 | 64.06 | 129.23 | 0.000 |
| 40 Oliveira 2017 | 51.10 | 34.64 | 63.41 | 126.79 | 0.000 |
| 41 Orgiles 2014 | 51.45 | 35.14 | 63.65 | 127.70 | 0.000 |
| 42 Ortega 2007 | 51.91 | 35.80 | 63.97 | 128.91 | 0.000 |
| 43 Panagiotakos 2008 | 51.90 | 35.79 | 63.97 | 128.90 | 0.000 |
| 44 Pati 2014 | 50.69 | 34.06 | 63.13 | 125.75 | 0.000 |
| 45 Peart 2011 | 51.08 | 34.61 | 63.40 | 126.73 | 0.000 |
| 46 Peltzer 2011 | 52.03 | 35.98 | 64.06 | 129.26 | 0.000 |
| 47 Pengpid 2018 | 52.06 | 36.02 | 64.08 | 129.34 | 0.000 |

*(Continued)*

**Table 3.** (Continued)

| Study omitted | I2 | [95% Conf. Interval] | | Chi2 | P>t |
|---|---|---|---|---|---|
| 48 Rani 2013 | 50.73 | 34.12 | 63.16 | 125.85 | 0.000 |
| 49 Rosi 2017 | 51.44 | 35.14 | 63.65 | 127.69 | 0.000 |
| 50 Saikia 2016 | 51.61 | 35.37 | 63.77 | 128.13 | 0.000 |
| 51 Savva 2002 | 52.06 | 36.02 | 64.08 | 129.33 | 0.000 |
| 52 Scanferla de Siqueira 2007 | 51.52 | 35.25 | 63.71 | 127.90 | 0.000 |
| 53 Shankaran 2011 | 52.00 | 35.93 | 64.04 | 129.16 | 0.000 |
| 54 Silva 2016 | 50.79 | 34.19 | 63.20 | 125.98 | 0.000 |
| 55 Silveira 2006 | 52.04 | 35.99 | 64.06 | 129.27 | 0.000 |
| 56 Teo 2014 | 50.64 | 33.99 | 63.10 | 125.62 | 0.000 |
| 57 Thibault 2010 | 47.25 | 29.11 | 60.75 | 117.54 | 0.000 |
| 58 Urrutia-Rojas 2008 | 51.94 | 35.85 | 64.00 | 129.02 | 0.000 |
| 59 Veugelers 2005 | 50.30 | 33.49 | 62.86 | 124.74 | 0.000 |
| 60 Watharkar 2015 | 51.41 | 35.08 | 63.63 | 127.59 | 0.000 |
| 61 Wethington 2013 | 51.72 | 35.53 | 63.84 | 128.42 | 0.000 |
| 62 Wilkie 2016 | 52.06 | 36.02 | 64.08 | 129.33 | 0.000 |
| 63 Winkvist 2016 | 51.58 | 35.33 | 63.74 | 128.04 | 0.000 |
| 64 Wittmeier 2008 | 51.89 | 35.77 | 63.96 | 128.86 | 0.000 |
| **Combined** | **51.29** | **35.06** | **63.46** | **129.34** | **0.000** |

heterogeneity. This 'one-out' approach tells us to what extent the overall heterogeneity changes by excluding a particular study at a time.

The Metaplot has a limitation. When the number of studies is very large (more than 35) as shown in Fig 3, the study codes in the x-axis come together and even may collapse due to space constraints. In such cases, the identification of the study codes may be difficult. Fortunately, the properties of the "metaplot" module solved this problem. In addition to the "Metaplot", this module generates a table in the "Results window" of the Stata and gives more details of 'one-out' sensitivity analysis including the $I^2$ and the $\chi^2$ statistics and their *P*-values as well as the studies codes and the studies identifications. Therefore, by turning back to the "Results window" we can realize which study has the greatest impact on the overall heterogeneity based on the $I^2$ and $\chi^2$ statistics.

## Conclusion

Metaplot is a visual complementary approach for testing between-study heterogeneity. This plot is a simple graphical approach that gives a quick and easy identification of the studies having substantial influences on overall heterogeneity as fast as possible. This method is based on 'one-out' sensitivity analysis and provides information both graphically and quantitatively about the extent of the overall heterogeneity changes by excluding a particular study at a time in terms of $I^2$ and $\chi^2$ statistics. It is possible to implement this graph for the meta-analysis of different types of outcome data.

## Supporting information

**S1 File.**
(ADO)

**S2 File.**
(HLP)

**S1 Dataset.**
(DTA)

**S2 Dataset.**
(DTA)

**S3 Dataset.**
(DTA)

# Author Contributions

**Conceptualization:** Jalal Poorolajal.

**Formal analysis:** Jalal Poorolajal.

**Investigation:** Jalal Poorolajal.

**Methodology:** Jalal Poorolajal, Shahla Noornejad.

**Project administration:** Jalal Poorolajal.

**Resources:** Jalal Poorolajal.

**Software:** Jalal Poorolajal, Shahla Noornejad.

**Supervision:** Jalal Poorolajal.

**Validation:** Jalal Poorolajal, Shahla Noornejad.

**Visualization:** Jalal Poorolajal.

**Writing – original draft:** Jalal Poorolajal.

**Writing – review & editing:** Shahla Noornejad.

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
