## [Decision Letter · Decision Letter 0]

14 Apr 2021

PONE-D-21-06251

Metaplot: A new Stata module for assessing heterogeneity in a meta-analysis

PLOS ONE

Dear Dr. Poorolajal,

Thank you for submitting your manuscript to PLOS ONE. After careful consideration, we feel that it has merit but does not fully meet PLOS ONE’s publication criteria as it currently stands. Therefore, we invite you to submit a revised version of the manuscript that addresses the points raised during the review process.

We look forward to receiving your revised manuscript.

Kind regards,

Mohammad Asghari Jafarabadi

Academic Editor

PLOS ONE

Journal Requirements:

Reviewers' comments:

Reviewer's Responses to Questions

**Comments to the Author**

1. Is the manuscript technically sound, and do the data support the conclusions?

Reviewer #1: Yes

2. Has the statistical analysis been performed appropriately and rigorously? 

Reviewer #1: Yes

3. Have the authors made all data underlying the findings in their manuscript fully available?

Reviewer #1: Yes

4. Is the manuscript presented in an intelligible fashion and written in standard English?

Reviewer #1: Yes

5. Review Comments to the Author

Reviewer #1: Author may add the reference to the work done by Huedo-Medina, T. B., Sánchez-Meca, J., Marín-Martínez, F., & Botella, J. (2006). Assessing heterogeneity in meta-analysis: Q statistic or I² index?. Psychological methods, 11(2), 193.”

The author has used illogically his 3 papers in the reference.

6. PLOS authors have the option to publish the peer review history of their article (what does this mean?). If published, this will include your full peer review and any attached files.

Reviewer #1: No

---

## [Author Response · Author response to Decision Letter 0]

15 Apr 2021

Editor’s comments

Answer: Done

Answer: Done

Answer: Done

Reviewers' comments:

Reviewer #1

Author may add the reference to the work done by Huedo-Medina, T. B., Sánchez-Meca, J., Marín-Martínez, F., & Botella, J. (2006). Assessing heterogeneity in meta-analysis: Q statistic or I² index?. Psychological methods, 11(2), 193.”

Answer: We added a paragraph in the discussion section and explain the results of this paper.

The author has used illogically his 3 papers in the reference.

Answer: You are right. The topics of these papers are not relevant to the topic of the current manuscript. However, we used real data from our previously published studies. We gave reference to these studies to indicate the reality of the data and their originality. However, if the honorable reviewers insist, we will remove these papers from the reference list.

Thank you.

---

## [Decision Letter · Decision Letter 1]

18 May 2021

PONE-D-21-06251R1

Metaplot: A new Stata module for assessing heterogeneity in a meta-analysis

PLOS ONE

Dear Dr. Poorolajal,

Thank you for submitting your manuscript to PLOS ONE. After careful consideration, we feel that it has merit but does not fully meet PLOS ONE’s publication criteria as it currently stands. Therefore, we invite you to submit a revised version of the manuscript that addresses the points raised during the review process.

We look forward to receiving your revised manuscript.

Kind regards,

Mohammad Asghari Jafarabadi

Academic Editor

PLOS ONE

Reviewers' comments:

Reviewer's Responses to Questions

**Comments to the Author**

1. If the authors have adequately addressed your comments raised in a previous round of review and you feel that this manuscript is now acceptable for publication, you may indicate that here to bypass the “Comments to the Author” section, enter your conflict of interest statement in the “Confidential to Editor” section, and submit your "Accept" recommendation.

Reviewer #1: All comments have been addressed

Reviewer #2: (No Response)

2. Is the manuscript technically sound, and do the data support the conclusions?

Reviewer #1: Yes

Reviewer #2: Yes

3. Has the statistical analysis been performed appropriately and rigorously? 

Reviewer #1: Yes

Reviewer #2: N/A

4. Have the authors made all data underlying the findings in their manuscript fully available?

Reviewer #1: Yes

Reviewer #2: Yes

5. Is the manuscript presented in an intelligible fashion and written in standard English?

Reviewer #1: Yes

Reviewer #2: Yes

6. Review Comments to the Author

Reviewer #1: (No Response)

Reviewer #2: The authors present a STATA module for assessing heterogeneity in a meta analysis using a "one out" sensitivity analysis. An advantage of this package is that it makes trivial what can be an extremely tedious process. This module likely has utility for some researchers conducting meta analyses and I believe that providing peer reviewed documentation for modules like these is good practice.

However, what the authors present is not a new module but appears to be a modification of an existing module (on which they published a paper in The Iranian Journal of Public Health in 2010). If I understand the manuscript correctly, the primary changes made to the old model were to modify plot generation so that it created a more interpretable figure and to include a table that summarized the relevant statistics of each iteration. The original module is not mentioned in the manuscript until halfway through the discussion section. If there are other key differences, they are not discussed in the manuscript.

I do not believe that this manuscript introduces sufficient new information (or introduces old information in a sufficiently more comprehensive or accessible manner) to warrant a full research article. The relevant new information introduced might be better suited as patch notes for the original module. If the authors wrote a different paper, one which focused on the changes made to the package and how those changes impact interpretation/conclusions, I could see that paper being more valuable.

Additional comments in attachment.

7. PLOS authors have the option to publish the peer review history of their article (what does this mean?). If published, this will include your full peer review and any attached files.

Reviewer #1: No

Reviewer #2: No

---

## [Author Response · Author response to Decision Letter 1]

18 May 2021

Reviewer comments: "Metaplot: A new Stata module for assessing heterogeneity in a meta-analysis" 

Sections quoted from the manuscript are in red. 

2021-05-07

• Title

Is this actually a new Stata module or an update of the older Metaplot module? If it is the latter, language should be changed accordingly. 

Answer: This Stata module is generated for the first time. The paper that we published in 2010 ONLY introduced the IDEA of Metaplot. But we did not generate the Stata module at that time. It took a long time until we could generate the module actually in the present form and ready for practice.

• Abstract

Results: first sentence presupposes that there is one study causing heterogeneity when that may or may not be the case. Consider instead: Metaplot allows rapid identification of studies that have a disproportionate impact on heterogeneity across studies, and communicates to what extent omission of that study may reduce the overall heterogeneity based on the I2 and χ2 statistics. 

Answer: We replaced the first sentence with the suggested sentence. Thank you.

Results: I'm not sure these are so much results as an assertion as to what the authors believe Metaplot can do. It might be better to overview the performance of Metaplot in the practical examples here.

Answer: This module was tested on real datasets with both “binomial” and “continuous” outcomes. Table 1 is an example of the real dataset with a “binomial” outcome (stomach cancer). Table 2 is another example of a real dataset with a “continuous” outcome (blood pressure). Table 2 is the third example of a real dataset with a binomial outcome (childhood obesity) with multiple studies. Therefore, this module can work in any situation (either limited or multiple studies) and with any outcome (either binomial or continuous). To clarify this ambiguity, we specified the type of outcomes in the results section. 

• Introduction

Your opening sentence is made a bit awkward by the inclusion of a non-defining clause where a defining clause should be used. Easily fixed by changing it to: The studies that are brought together in a meta-analysis inevitably differ in many aspects.

Answer: We replaced the opening sentence with the suggested sentence. Thank you.

• Methods

Potential typo at the bottom of page 3. Language is referred to as "Meta" but functions are called "Mata" functions. I'm not a Stata user so please ignore this comment if my understanding is incorrect. 

Answer: The correct form of the word is “Mata”. Thank you.

Is the first paragraph introducing mata functions even necessary given that you don't use or discuss this information later in the paper? 

Answer: You are right. It is not necessary. Therefore, we deleted the first paragraph from the methods section.

• Discussion

Performing sensitivity analyses based on the sequential and combinatorial algorithm proposed by Patsopoulos et al [5]. This is a sentence fragment and should be corrected. 

Answer: We corrected the sentence as follows: “Patsopoulos et al [5] suggested the sequential and combinatorial algorithm for performing sensitivity analyses.”

"boring," subjective judgement. "Time consuming" is sufficient. Consider exclusion. 

Answer: The word “boring’ was removed.

Although “metaplot” was first introduced in 2010[10], however, it was a preliminary idea that changed a lot over time. "However" can be removed here. 

Answer: The word “however’ was removed.

Although “metaplot” was first introduced in 2010[10], however, it was a preliminary idea that changed a lot over time. The new design of the “metaplot” presented in this paper is very different from the original one introduced in 2010. The original design was a complicated three-dimensional graph with x, y, and z axes including unnecessary information. It was rather hard to understand. The new design of “metaplot” is a two-dimensional graph with x and y axes. Furthermore, we added a table including details of information (I2 and χ2 statistics and their P-values omitting one study in each turn) to simplify the interpretation of the ‘metaplot’ graph. 

So what this paper is actually introducing is a modification to an existing package? If I'm understanding this paragraph correctly, the updated package is using fundamentally the same methodology but provides a changed graphical output and additional tables that improve ease of interpretation. While I support providing peer reviewed documentation for statistical packages that can be cited in papers that use the package, I don't think that this manuscript introduces sufficient new information (or introduces old information in a sufficiently more comprehensive or accessible manner) to warrant a full research article. This feels like something that could be attached to the patch notes for the package. 

Answer: In the paper that we published in 2010, we only introduced the raw idea of “Metaplot”. We never generated a package for practice. After the publication of the paper, several researchers from around the world contacted me and requested to submit the Stata module for them, but I apologized to them because there was no Stata module at that time. This idea was in my mind until now (after ten years) that I could find an expert software engineer (my coauthor Shahla) who helped me to generate the Stata module. During this process, we changed the preliminary idea and improved its potential capability in the present form. In 2010, we prepared the figure manually by Excel software to present our subjective idea. The appearance of the figure introduced in 2010 is completely different from what we produced now by the Stata module. In the current paper, we explained the capability of the “Metaplot” module and how to use the Stata command and its options. We examined this module, which we generated recently, on different real datasets and reported the results in Tables 1-3 and Figures 1-3. We added an explanation to the discussion section to clarify this ambiguity.

Reviewer #2: 

The authors present a STATA module for assessing heterogeneity in a meta analysis using a "one out" sensitivity analysis. An advantage of this package is that it makes trivial what can be an extremely tedious process. This module likely has utility for some researchers conducting meta analyses and I believe that providing peer reviewed documentation for modules like these is good practice.

However, what the authors present is not a new module but appears to be a modification of an existing module (on which they published a paper in The Iranian Journal of Public Health in 2010). If I understand the manuscript correctly, the primary changes made to the old model were to modify plot generation so that it created a more interpretable figure and to include a table that summarized the relevant statistics of each iteration. The original module is not mentioned in the manuscript until halfway through the discussion section. If there are other key differences, they are not discussed in the manuscript.

I do not believe that this manuscript introduces sufficient new information (or introduces old information in a sufficiently more comprehensive or accessible manner) to warrant a full research article. The relevant new information introduced might be better suited as patch notes for the original module. If the authors wrote a different paper, one which focused on the changes made to the package and how those changes impact interpretation/conclusions, I could see that paper being more valuable.

Answer: In the paper that we published in 2010, we only introduced the raw idea of “Metaplot”. We never generated a package for practice. After the publication of the paper, several researchers from around the world contacted me and requested to submit the Stata module for them, but I apologized to them because there was no Stata module at that time. This idea was in my mind until now (after ten years) that I could find an expert software engineer (my coauthor Shahla) who helped me to generate the Stata module. During this process, we changed the preliminary idea and improved its potential capability in the present form. In 2010, we prepared the figure manually by Excel software to present our subjective idea. The appearance of the figure introduced in 2010 is completely different from what we produced now by the Stata module. In the current paper, we explained the capability of the “Metaplot” module and how to use the Stata command and its options. We examined this module, which we generated recently, on different real datasets and reported the results in Tables 1-3 and Figures 1-3. We added an explanation to the discussion section to clarify this ambiguity.

---

## [Decision Letter · Decision Letter 2]

3 Jun 2021

Metaplot: A new Stata module for assessing heterogeneity in a meta-analysis

PONE-D-21-06251R2

Dear Dr. Poorolajal,

We’re pleased to inform you that your manuscript has been judged scientifically suitable for publication and will be formally accepted for publication once it meets all outstanding technical requirements.

Kind regards,

Mohammad Asghari Jafarabadi

Academic Editor

PLOS ONE

Additional Editor Comments (optional):

The references should be set according to the journal style using a reference manager. 

Reviewers' comments:

Reviewer's Responses to Questions

**Comments to the Author**

1. If the authors have adequately addressed your comments raised in a previous round of review and you feel that this manuscript is now acceptable for publication, you may indicate that here to bypass the “Comments to the Author” section, enter your conflict of interest statement in the “Confidential to Editor” section, and submit your "Accept" recommendation.

Reviewer #3: All comments have been addressed

2. Is the manuscript technically sound, and do the data support the conclusions?

Reviewer #3: Partly

3. Has the statistical analysis been performed appropriately and rigorously? 

Reviewer #3: Yes

4. Have the authors made all data underlying the findings in their manuscript fully available?

Reviewer #3: Yes

5. Is the manuscript presented in an intelligible fashion and written in standard English?

Reviewer #3: Yes

6. Review Comments to the Author

Reviewer #3: Everything is okayو The method of writing the research is good and the way to describe the results is good, but it needs to arrange the sources in one of the source ranking programs such as Mendeley

7. PLOS authors have the option to publish the peer review history of their article (what does this mean?). If published, this will include your full peer review and any attached files.

Reviewer #3: No

---

## [Editor Report · Acceptance letter]

17 Jun 2021

PONE-D-21-06251R2 

Metaplot: A new Stata module for assessing heterogeneity in a meta-analysis 

Dear Dr. Poorolajal:

I'm pleased to inform you that your manuscript has been deemed suitable for publication in PLOS ONE. Congratulations! Your manuscript is now with our production department. 

Kind regards, 

on behalf of

Professor Mohammad Asghari Jafarabadi 

Academic Editor

PLOS ONE